# Effects of Acid-Anhydride-Modified Cellulose Nanofiber on Poly(Lactic Acid) Composite Films

**DOI:** 10.3390/nano11030753

**Published:** 2021-03-17

**Authors:** Naharullah Jamaluddin, Yu-I Hsu, Taka-Aki Asoh, Hiroshi Uyama

**Affiliations:** Department of Applied Chemistry, Graduate School of Engineering, Osaka University, 2-1 Yamadaoka, Suita, Osaka 565-0871, Japan; naharullah@chem.eng.osaka-u.ac.jp (N.J.); asoh@chem.eng.osaka-u.ac.jp (T.-A.A.)

**Keywords:** cellulose, nanofiber, acid anhydrides, poly(lactic acid), nanocomposite, optical properties, mechanical properties

## Abstract

In this study, we investigated the effect of the addition of cellulose nanofiber (CNF) fillers on the performance of poly(lactic acid) (PLA). Modification of the hydroxyl group of cellulose to the acyl group by acid anhydrides changed the compatibility of the CNF with PLA. CNF was modified by acetic anhydride, propionic anhydride, and butyric anhydride to form surface-modified acetylated CNF (CNFa), propionylated CNF (CNFp), and butyrylated CNF (CNFb), respectively, to improve the compatibility with the PLA matrix. The effects of the different acid anhydrides were compared based on their rates of reaction in the acylation process. PLA with modified cellulose nanofiber fillers formed smoother surfaces with better transparency, mechanical, and wettability properties compared with the PLA/CNF composite film. The effects of CNFa, CNFp, and CNFb on the PLA matrix were compared, and it was found that CNFp was the best filler for PLA.

## 1. Introduction

Bio-based polymers are attracting attention as an alternative to existing materials such as synthetic plastics and petroleum-based materials. The development of bio-based polymers is the main aspect of maintaining a sustainable society because of their renewability and biocompatibility, which is parallel with one of the 17 sustainable development goals. Some plastics can be decomposed by combustion, such as polyhydroxyalkanoate, which is widely applied in tissue engineering applications, and poly(lactic acid) (PLA), which is produced by fermentation of corn or sugarcane [1,2]. PLA has interesting features, such as good processability and transparency [3]. In industry, PLA is currently commercialized as single-use disposal packaging [4]. However, PLA also has several drawbacks that limit its applications, such as low thermal, mechanical, and barrier properties [5]. Therefore, a lot of research has been carried out to enhance and counter these drawbacks by preparing PLA composites [6]. The combination of PLA and cellulose as a nanocomposite is expected to exhibit improved properties by combination of the plastic matrix and cellulose filler. Several types of research have been performed to produce PLA/cellulose composites using different processing methods. In 2008, Alemdar and Sain [7] investigated the structure and thermal properties of PLA/cellulose-whisker nanocomposites by solution casting. In 2007, Iwamoto et al. [8] fabricated a PLA/cellulose-nanofiber (CNF) composite by the extrusion mixing method. Modification of cellulose before composite preparation with PLA has also been widely investigated, and most of the modifications improved dispersion of cellulose in the PLA matrix [9].

In each d-glucose unit of cellulose, there are three active hydroxyl groups. These groups can easily form hydrogen bonds, resulting in cellulose showing hydrophilicity. Cellulose is insoluble in common organic solvents and hydrophobic polymers [10]. After modification of the hydroxyl groups of cellulose, the hydrogen bonds are weakened, and cellulose becomes more dispersible in organic solvents and most hydrophobic polymers [11]. In addition, cellulose is one of the best filler materials for polymer composites because of its high mechanical strength and biodegradability, and several functionalizations of cellulose have been studied and developed [12]. Fujisawa et al. [13] reported ionic exchange of (2,2,6,6-tetramethylpiperidin-1-yl)oxyl-oxidized CNF by grafting it with poly(ethylene glycol) using organic solvents such as chloroform, toluene, and tetrahydrofuran. Other modifications of cellulose include esterification of bacterial cellulose (BC), alkylation of micro- and nanocellulose, silanization by 3-methacrysloxypropyltrimethoxysilane, and glyoxalization of BC networks [14,15,16,17]. Various modifications of cellulose have been performed to improve the compatibility of cellulose with the polymer matrix [18,19]. Recently, our group successfully modified cellulose with citric acid to improve its compatibility with poly(propylene) and PLA resins [20,21]. Furthermore, esterification of cellulose to produce hydrophobic products has been widely studied since the early 1980s [22].

The dependence of the esterification effects of CNF on the starting reactants and their degrees of substitutions (DSs) is also an interesting topic. For example, Vice-Garcia et al. [23] investigated the dependence of the main transition of cellulose fatty esters on the length of their aliphatic substituents. In 2009, Crépy et al. [24] investigated the effect of saturated and unsaturated C12 to C18 chains on cellulose fatty esters and prepared polymer films from modified cellulose. Recently, the same group studied a series of fatty acid cellulose esters (FACEs) with various DS values and side-chain lengths from C10 to C16, and they compared the mechanical and chemical properties of each FACE [25]. A review of the various material functionalities based on thermoplastic cellulose and related structural polysaccharide derivatives has been published [26]. In the review, they discussed the approaches for enabling effective thermoplasticization and incorporation of material functionalities, such as single-substituent derivatization, derivatization with multiple substituents, blending of simple derivatives, and graft copolymerization. The influences of short alkyl-chain substituents have also been investigated. For example, Yu et al. [27] modified cellulose with acid anhydrides to form modified cellulose with C1 to C3 alkyl-chain substituents, and then investigated their effects on the structure and thermal properties of cellulose-g-polyoxyethylene (2) hexadecyl ether. However, very limited research on comparison of the effects of different alkyl-chain substituents of CNF on PLA composite films has been performed.

In this work, CNF was modified with acetic anhydride (AA), propionic anhydride (PA), and butyric anhydride (BA) to form surface-modified acetylated CNF (CNFa), propionylated CNF (CNFp), and butyrylated CNF (CNFb), respectively. The acylation method of the CNF was developed based on our previous research [11]. Fillers of modified CNFs (m-CNFs) with different DS values and lengths of the alkyl-chain substituents were used to fabricate PLA/m-CNF composite films with various transparency, strength, and wettability values. The aims of this work are to investigate the effect of the acid anhydride on the rate of CNF acylation and the influence of the acylated CNF on the PLA/m-CNF composite film. The acylation reaction time was varied to obtain m-CNFs with various DS values and predict the rate of acylation. CNFa, CNFp, and CNFb differ in the length of the alkyl-chain substituents of the carbonyl groups (Figure 1), which varies from C1 to C3. The PLA/m-CNF composites were expected to improve the mechanical and wettability properties. We focused on the dispersibility and optical, physical, and hydrophobic properties of the PLA/m-CNF composite films measured by ultraviolet–visible (UV–vis) spectroscopy, a haze meter, scanning electron microscopy (SEM), tensile strength tests, and water contact angle (WCA) measurements.

## 2. Materials and Methods

### 2.1. Materials

Microcrystalline cellulose (MCC) was obtained from Merck Japan Ltd. (Tokyo, Japan). *N*,*N*-dimethylformamide (DMF) was purchased from Kanto Chemical (Tokyo, Japan). AA, PA, BA, chloroform (>99.0%), and acetone (>99.0%) were obtained from Wako Pure Chemical Industries (Osaka, Japan) and used without further treatment. PLA (PLA2003D Ingeo biopolymer) was purchased from NatureWorks LLC (Minnesota, MN, USA). Water was treated with a Model III instrument from Organo Corporation (Tokyo, Japan) to produce deionized water (H_2_O).

### 2.2. Preparation of CNF

Fibrillation of MCC was performed with a stone grinding machine (Masuko Sangyo, Saitama, Japan) without other chemical treatments. In brief, MCC (40 g) was soaked and stirred in water for 2 days to produce a suspension of 2 wt% MCC. The MCC suspension was then ground at 1500 rpm for 10 cycles with a stone grinder grit size of 80 (ultra-fine). As the fiber size decreased, MCC automatically exited the chute in the form of CNF. Each cycle was repeated when the hopper was almost empty by transferring the product of the previous cycle back into the hopper. After 10 cycles, the suspension of CNF (1.57 wt%) in water was stored in a glass bottle and kept in a refrigerator (<4 °C).

### 2.3. Surface Modification of CNF

First, 0.81 g of CNF in water (1.57 wt%) was homogenized with 80 mL of DMF. The CNF suspension in this mixed solvent was transferred to a rotary evaporator to remove the water from the system. The mixture was then transferred into a round-bottomed flask connected to a reflux condenser on a hotplate heated to 110 °C using silicon oil, and simultaneously 0.5 mol of AA, PA, or BA was added. The acylation reaction time varied from 1 to 4 h to investigate the reaction rate. The solution was then quenched in an ice bath followed by addition of acetone (50 mL). The solution was then centrifuged and washed several times with acetone to remove the unreacted chemicals and DMF. Finally, the medium was exchanged with chloroform to obtain the m-CNF in chloroform. The products were stored in a refrigerator (<4 °C) before preparation of the PLA/m-CNF composite films. The products of the m-CNFs are called CNFa*X* (AA modified), CNFp*X* (PA modified), and CNFb*X* (BA modified) based on their anhydride, where *X* is the reaction time (in h).

### 2.4. Preparation of PLA/m-CNF Composites

The solution casting method was used to prepare the PLA/m-CNF composites. First, a dispersion of unmodified CNF or m-CNF in chloroform (0.02 g of solid) was poured into a beaker, and chloroform was added to reach 50 g in total weight. Next, 2 g of PLA was added, and the mixture was stirred for 3 h at room temperature. The mixture was then poured into a Petri dish and left overnight in a closed bio-shaker at 25 rpm and 40 °C to evaporate the chloroform. The samples are denoted neat PLA (without filler), PLA/CNF, PLA/CNFa, PLA/CNFp, and PLA/CNFb composite films.

### 2.5. Characterization

An attenuated total reflection infrared (ATR-IR) spectrometer (iD5 ATR, Thermo Scientific, Waltham, MA, USA) was used to determine the functional groups after modification. Electron-dispersive X-ray (EDX) spectroscopy was carried out to calculate the DS values of the produced m-CNFs using a Miniscope TM3000/SwiftED3000 system (Hitachi, Tokyo, Japan). For the DS calculation, C and O, but not H, were included. After acylation, the mass ratio of C is expected to increase relative to O owing to the increased C species from the acyl groups that substitute for the H atoms of the hydroxyl groups of CNF. The transmittance and transparency values of the composite films were determined using a UV–vis spectrophotometer (U-2810, Hitachi, Tokyo, Japan) in the visible region (200–800 nm) scanned at a rate of 800 nm/min. The transparency values were calculated by
(1)$$\mathrm{Transparency}\;=\log\;\frac{T\left(\mathrm{average}\right)}{X\;}$$
where *T*(average) and *X* are the average UV–vis transmittance and average thickness of the film, respectively. A haze meter (NDH 4000, Nippon Denshoku) was used to calculate the haze values of the PLA/m-CNF composite films. The haze value was determined by
(2)$$\mathrm{Haze}=\frac{T.T.-P.T.}{T.T.\;}\times 100$$
where T.T. and P.T. are the intensities of the transmitted light and parallel light, respectively. For the transmittance studies, two batches of PLA/m-CNF composite films with different thicknesses were fabricated. The thinner films were prepared by reducing the total amount of reactant to 1 g while maintaining their ratios. The morphologies of the PLA/m-CNF composite films were observed by SEM (SU3500, Hitachi, Tokyo, Japan) using Au–Pd sputter to increase the sample’s surface conductivity. The samples were cut and attached to circular SEM plates (25 cm). For the surface morphology, magnification at 100× (500 μm scale) was used to provide a wide area of the film’s surface. The mechanical properties of the composite films were investigated with a universal testing machine (EZ Graph, Shimadzu, Kyoto, Japan) following the JIS K6251-8 standard. Each sample was cut into dumbbell shapes for at least five tests and dried in an oven at 80 °C before analysis to remove any remaining solvent and moisture. The crosshead speed was 50 mm/min with a load cell of 100 N. The crystallinities of PLA/m-CNF composite films were calculated using differential scanning calorimetry (DSC) (EXSTAR 6000 DSC6220, SEIKO, Tokyo, Japan) from crystallization temperature (T_c_) and melting temperature (T_m_) areas. The crystallinities of the PLA composite films were calculated using the theoretical heat of fusion (∆H_f_) (93.1 J g^−1^) [28].
(3)$$\Delta H_{m}-\Delta H_{c}=\;\Delta H^{\prime }$$
(4)$$\frac{\Delta H^{\prime }}{\Delta H_{f}}\times \;100=\;X_{c}\;\%$$
∆H_f_ = 93.1 J/g
where ∆H_m_ and ∆H_c_ are the enthalpies given during melting and crystallization, respectively, and X_c_ % is the degree of crystallinity. The wettability of the films was investigated by WCA measurement using a Drop Master DM300 contact angle meter (Kyowa Interface Science, Tokyo, Japan) with FAMAS basic software. Each sample was dried in an oven at 80 °C before analysis. The WCAs were measured for at least seven specimens, and the four values with the least deviation were used to calculate the average WCA.

## 3. Results and Discussion

### 3.1. Surface Modification of CNF

ATR-IR was performed to observe the acyl groups that substituted the hydroxyl groups of CNF after modification. The ATR-IR spectra of the m-CNFs are shown in Figure 2. New peaks appeared at around 1730 cm^−1^, corresponding to the C=O stretching vibration modes of the carbonyl group between CNF and the anhydrides. This indicates that the acyl groups from the anhydrides were incorporated into the CNF to become CNFa, CNFp, and CNFb. Moreover, the increases in the intensities of the carbonyl peaks with increasing reaction time showed that the acylation processes were proportional to the reaction time. The intensities of the –OH deformation of water peaks at around 1650 cm^−1^ were higher than the intensities of the C=O stretching peaks for all of the m-CNFs after 1-h acylation. This changed when the acylation time was increased to 2 h, where the intensities of the carbonyl peaks were higher than those of the –OH deformation of water peaks. Furthermore, the intensities of the C=O stretching peaks were higher for 4-h acylation compared with both 1- and 2-h acylation, indicating that the DS for 4-h acylation was the highest among the studied reaction times.

### 3.2. Degree of Substitution

EDX spectroscopy has a probe depth of 1–3 µm, while the average diameter of the CNF was about 63 ± 17 nm. The average diameter was calculated based on the SEM image in Figure 3. EDX indicates the composition of each element, and therefore the DS values of the CNF and m-CNFs can be calculated. The modification percentages of the m-CNFs with respect to the reaction time are shown on Figure 4. The modification percentages increased with increasing reaction time from 1 to 4 h. The rates of acylation of CNFa and CNFp were almost the same and exponentially increased with the reaction time, while CNFp showed a slightly higher acylation rate. However, the acylation rate of CNFb showed that the acylation rate was significantly lower for longer alkyl-chain substituent. Longer alkyl-chain of the anhydride (BA) leads to a slower rate reaction due to steric effect and lower surface area compared to AA and PA [29]. After 1-h reaction, only 9.8% of CNFb was modified, which was the lowest among the m-CNFs. For 4 h reaction time, 24.5% of CNFb was modified, while 29.4% and 31.4% of CNFa and CNFp were modified, respectively. The EDX results further verified that the acyl groups were chemically bonded to the CNF and acylation increased with increasing reaction time. The DS values of the m-CNFs were calculated from the percentage modified values using Equation (S4), and the results are given in Table 1 and Appendix A.

### 3.3. Optical Transmittance of the PLA/m-CNF Composite Films

Photographs of the PLA/m-CNF composite films are shown in Figure 5. The patterns in the background can be clearly observed through the neat PLA film, demonstrating that the film possessed very good transparency (Figure 5a). The white agglomerates observed on the PLA/CNF composite film (Figure 5b) are because of its inhomogeneity with the PLA matrix. CNF tends to form agglomerates in the matrix because of their opposite hydrophilicity. Agglomerates of CNFa2, CNFp2, CNFa4, and CNFp4 in the PLA matrix were confirmed to be minimal because the background can still be clearly observed without any precipitation of the fillers. Even though the PLA/CNFb4 composite film (Figure 5h) exhibited high transparency because the background can be observed, the lower DS of CNFb2 (PLA/CNFb2 composite film) (Figure 5g) resulted in poor visibility of the background. This might be because the lower DS of CNFb2 (0.47) makes the unmodified CNF (in CNFb2) have a more significant effect on the PLA/m-CNF composite film, and hence reduces its miscibility with the PLA matrix. In addition, the surfaces of the films with the CNFb filler were rougher those with the CNFa and CNFp fillers regardless of their DS. As previously mentioned, the m-CNFs (DS > 0.47) have high compatibility with PLA because of their similar hydrophobicity. For this reason, the PLA/CNFa, PLA/CNFp, and PLA/CNFb4 composite films showed better transparency than the PLA/CNF and PLA/CNFb2 composite films.

### 3.4. Light Transmittance of the PLA/m-CNF Composite Films

UV–vis spectroscopy was performed to compare the transparency of the PLA/m-CNF composite films by calculating the percentage of UV–vis transmittance. We compared two batches of PLA/m-CNF composite films with different thicknesses. The different thickness films were prepared by using different initial amounts of reactant (1 and 2 g) during preparation of the films. The average transmittance values (Table 2) were used as relative values for comparison. Thicker films show lower transparency because the amount of light that passes through the films is reduced. All of the films prepared with a larger amount of reactant showed UV–vis transmittance below 42.2%, including the neat PLA film. Furthermore, additional filler also decreases the transparency because the light is scattered/diffracted by the filler regardless of the thickness. Even though there was a very large difference between the UV–vis transmittance of the thick and thin films, both batches of PLA/m-CNF composite films showed the same pattern with respect to the filler. For 1 g of the reactant, the neat PLA film showed 89.3% transmittance, which was the highest among all of the films. The heterogeneous nature of the PLA/CNF composite film decreased the transmittance to 77.8%. This is lower than that of the PLA film because of formation of agglomerates, which result in diffraction and scattering of light during UV–vis analysis. PLA with the CNFb2 filler (DS = 0.47) also showed lower transmittance (77.9%) than the neat PLA film (89.3%) owing to the low compatibility between the filler and PLA. The composite films of PLA with the other m-CNF fillers showed better transmittance (81.0–83.9%). This can be attributed to the better and more uniform dispersion of the relatively high DS (≥0.74) m-CNFs in the PLA matrix compared with unmodified CNF or low DS m-CNFs. The transmittance spectra of the PLA/m-CNF composite films (1 g) in the visible wavelength region (400–800 nm) are shown in Figure 6, which were used to calculate the transparency values of the films. The transparency values were calculated based on the specific thickness of each film. All of the transparency values followed the same pattern as the UV–vis transmittance regardless of the amount of reactant used.

### 3.5. Haze Transmittance of the PLA/m-CNF Composite Films

To further support the UV–vis transmittance, the haze values of the films were determined (Table 3). The haze values increased with addition of the fillers. The neat PLA film gave the lowest haze value of 6.8%. The poor dispersions of CNF and CNFb2 showed haze values of 35.5% and 30.0%, respectively. Adding the m-CNF fillers changed the haze values to between 17.5% and 19.5% (except for CNFb2), which can be considered to be good for transparent films. The increase in the haze value from that of the neat PLA is because the fillers increased the amount of reflected and scattered light. The compatibility and dispersion of the higher DS m-CNF fillers in the PLA matrix are the key factors for achieving lower haze values compared with the PLA/CNF and PLA/CNFb2 composite films.

### 3.6. Morphologies of the PLA/m-CNF Composite Films

The CNF filler resulted in different physical properties of the PLA matrix in comparison with the m-CNFs fillers, especially on the surface, because of the difference in their compatibility. The PLA film (Figure 7a) showed a smooth surface because there was no filler involved. The PLA/CNF composite film had a rough surface (Figure 7b), and it was clear that the CNF filler caused agglomeration on/in the PLA matrix. Hence, the PLA/CNF composite film showed a lumpy surface with clear white agglomerates. In contrast, the PLA/m-CNF composite films with CNFa4 and CNFp4 as fillers (Figure 7d,f) exhibited flat and smooth surfaces because of their good compatibility with the PLA matrix. These results demonstrate that acylation changed the compatibility of the CNF with the PLA matrix for CNFa and CNFp. The relatively high DS m-CNF fillers were expected to have high compatibility with PLA owing to their hydrophobicity, while the hydrophilic CNF and relatively low DS m-CNFs were expected to have low compatibility with PLA. The SEM images showed that the scattered fillers on the surface were related to the DS of the m-CNF fillers. As the DS of the m-CNF decreased, the filler became more observable, as shown in Figure 7c,e. This indicates that the compatibility between the PLA matrix and m-CNF filler is affected by the difference in the DS values. As discussed in Section 3.3, the surfaces of the PLA/CNFb composite films were both rough. The SEM images of the PLA/CNFb composite films (Figure 7g,h) are in agreement with those observations, because the PLA/CNFb composite films were rough. The DS of CNFb4 is 0.74, which is almost the same as that of CNFa2 (DS = 0.75), so the difference in the lengths of the alkyl-chain substituents might be the cause of this opposite observation. It seems that the butyryl group substituted on the hydroxyl group of CNF tends to shield and localize CNFb, and hence rough surfaces are observed for the PLA/CNFb composite films regardless of their DS. Morphology differences of the m-CNFs are observed in the SEM images (Appendix A). After modification with acid anhydrides, the structures of the m-CNFs changed, and higher DS m-CNFs produced smaller agglomerates. CNFb4 (DS = 0.74) formed relatively large agglomerates, which might lead to a rougher surface during PLA/CNFb composite film preparation. It appears that the higher number of the hydroxyl groups in CNFb4 (due to lower DS than CNFa4 and CNFp4) formed an opposite interaction between the unmodified and modified parts of CNFb4; hence, larger agglomerates were formed during dispersion in chloroform.

Other than surface morphologies, the cross-section of the films also was observed (Figure 8). It is interesting to note that neat PLA and PLA/CNF composite films display plastic-like deformation after cracked (Figure 8a,b). It was proven that CNF is not compatible with the PLA matrix. Meanwhile, Figure 8c–h show that the m-CNFs are incorporated with the PLA matrix by forming stack-like morphologies.

### 3.7. Mechanical Properties of the PLA/m-CNF Composite Films

The mechanical properties of the films are given in Table 4. The neat PLA film showed a tensile strength of 46.1 ± 4.5 MPa, and inclusion of CNF in the PLA matrix decreased the tensile strength (23.9 ± 5.8 MPa). The decrease in the tensile strength of the PLA/CNF composite film was because of the inhomogeneity between the PLA matrix and CNF filler. The DS of the m-CNF played an important role in determining the tensile strength of the PLA/m-CNF composite film. Regardless of the filler species, all of the films with fillers of DS < 0.76 (CNFa2, CNFb2, and CNFb4) exhibited lower tensile strength than the neat PLA film. Conversely, the fillers with DS > 0.80 showed higher tensile strength, with the PLA/CNFp4 composite film showing the highest tensile strength of 53.0 ± 2.4 MPa. Both the CNFa4 (DS = 0.88) and CNFp2 (DS = 0.81) fillers also improved the tensile strength of the PLA/m-CNF composite films compared with the neat PLA film. Therefore, the tensile strength of the PLA film can be improved by addition of a m-CNF with DS > 0.80. In general, composites prepared by the solution casting method have lower tensile strength than those prepared by the melt blend method [30]. This also occurred for our products because the tensile strength of PLA was lower than that of pure commercialized or industrial PLA. The neat PLA film showed a Young’s modulus of 1.24 ± 0.28 GPa. The differences among the Young’s modulus values of the PLA/m-CNF composite films showed the same pattern as the tensile strength, except for the CNFa fillers. Even though the PLA matrix with the CNFa2 filler showed lower tensile strength than the neat PLA film (by 11%), its Young’s modulus was about 14% higher. In contrast, the PLA/CNFa4 composite showed higher tensile strength than the neat PLA film, but its Young’s modulus was lower (1.07 ± 0.10 GPa). Regarding the strain to failure, there were no significant differences among the strain percentages of the PLA/m-CNF composite films (2.4–3.9% with standard deviations of 0.1–1.4). The areas under the strain–stress curves (Appendix A) were calculated to predict the tensile toughness values of the films. Although the PLA/m-CNF composite films showed good tensile strength and modulus, their toughness was relatively poor, the same as general PLA [31].

### 3.8. Crystallinity of the PLA/m-CNF Composite Films

Both transparency and mechanical properties of PLA can be affected by the crystallinity of the samples [32]. Calculated from DSC thermograms (Appendix A), the X_c_ of PLA/m-CNF composite films is obtained and reported in Table 5. X_c_ of PLA/m-CNF composite films was lower compared to neat PLA (39.6%) due to the formation of aggregates, which decreased the number of nucleating sites [33]. PLA/m-CNF composite films showed that no significant variation was observed regardless of the m-CNFs’ substituent groups, which vary from 30.7% to 33.1%. Meanwhile, PLA/CNF composite film showed the most amorphous structure at X_c_ of 16.2% due to the incompatibility between CNF filler and PLA matrix, hence forming more aggregates than m-CNFs fillers. Therefore, PLA/CNF composite films exhibited a lower amount of nucleating sites.

### 3.9. Wettability of the PLA/m-CNF Composite Films

The WCAs of the neat PLA and PLA/m-CNF composite films are shown in Figure 9. The neat PLA film showed an average WCA of 83.9°, whereas that of the PLA/CNF composite film was 81.7°. This shows that the PLA/CNF filler had higher wettability owing to the hydrophilicity of CNF. All of the PLA/m-CNF composite films showed higher WCAs than the neat PLA and PLA/CNF films regardless of the filler and DS. This is attributed to a change from hydrophilic CNF to hydrophobic CNFa, CNFp, and CNFb. Substitution of the acyl groups for the hydroxyl groups of CNF is the main reason for the hydrophobicity. From the discussion in Section 3.1, the number of hydrogen bonds of CNF decreases when the hydroxyl groups are substituted by hydrophobic acyl groups. Therefore, the wettability of the PLA film also decreases by adding m-CNF fillers compared with unmodified CNF. Based on the results in Figure 9, the length of the alkyl-chain substituent had a significant effect on the wettability of the PLA/m-CNF composite film. Comparing the fillers with the same acylation time, PLA/CNFb showed the highest WCA values for both reaction times (87.3° and 88.5° for the CNFb2 and CNFb4 fillers, respectively). This indicates that the substituted propionyl group of CNFb increased the hydrophobicity of the PLA/m-CNF composite film even though the DS was lower than those of both CNFa and CNFp. Comparison between CNFa and CNFp also showed that the CNFa fillers had lower hydrophobicity (WCAs of 85.3° and 86.0° for CNFa2 and CNFa4, respectively) owing to their shorter alkyl-chain substituents.

## 4. Conclusions

In this study, CNF was acylated using acid anhydrides with short alkyl-chain substituents (C1 to C3), followed by fabrication of PLA/m-CNF composite films. CNFp showed the highest rate of acylation among the m-CNFs, and reaction for 4 h produced 31.4% CNFp, which was the highest among all of the studied parameters. The introduction of CNF as a filler decreased the transparency, mechanical, and wettability properties compared with the neat PLA film. Conversely, the compatibility of the m-CNFs with the PLA matrix mostly improved. By comparison of the films, the effects of different m-CNFs on PLA were investigated. The PLA/CNFp4 composite films showed the best transparency and mechanical properties, which was mainly because CNFp4 had the highest DS. Therefore, the DS of the filler is the key factor to improve the properties of PLA/m-CNF composite films. Meanwhile, the low DS CNFb2 filler (DS = 0.47) showed inhomogeneity of the matrix–filler interaction because the unmodified CNF part of CNFb2 had a more significant effect on the composite. Thus, the transparency and mechanical properties of the PLA/CNFb2 composite film were lower than those of the other PLA/m-CNF films. However, according to the WCA, the wettability of the PLA/CNFb2 composite film improved. Based on these observations, it was concluded that longer alkyl-chain substituent (CNFb filler) had a greater influence than the DS of the filler on the wettability properties of the PLA/m-CNF composite films.

## Figures and Tables

**Figure 1 nanomaterials-11-00753-f001:**
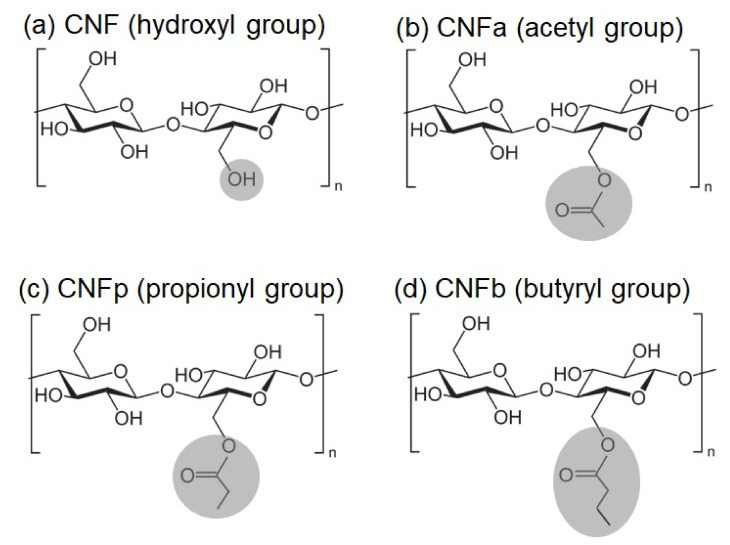
Chemical structures of (**a**) cellulose nanofiber (CNF); (**b**) surface-modified acetylated CNF (CNFa); (**c**) propionylated CNF (CNFp); and (**d**) butyrylated CNF (CNFb).

**Figure 2 nanomaterials-11-00753-f002:**
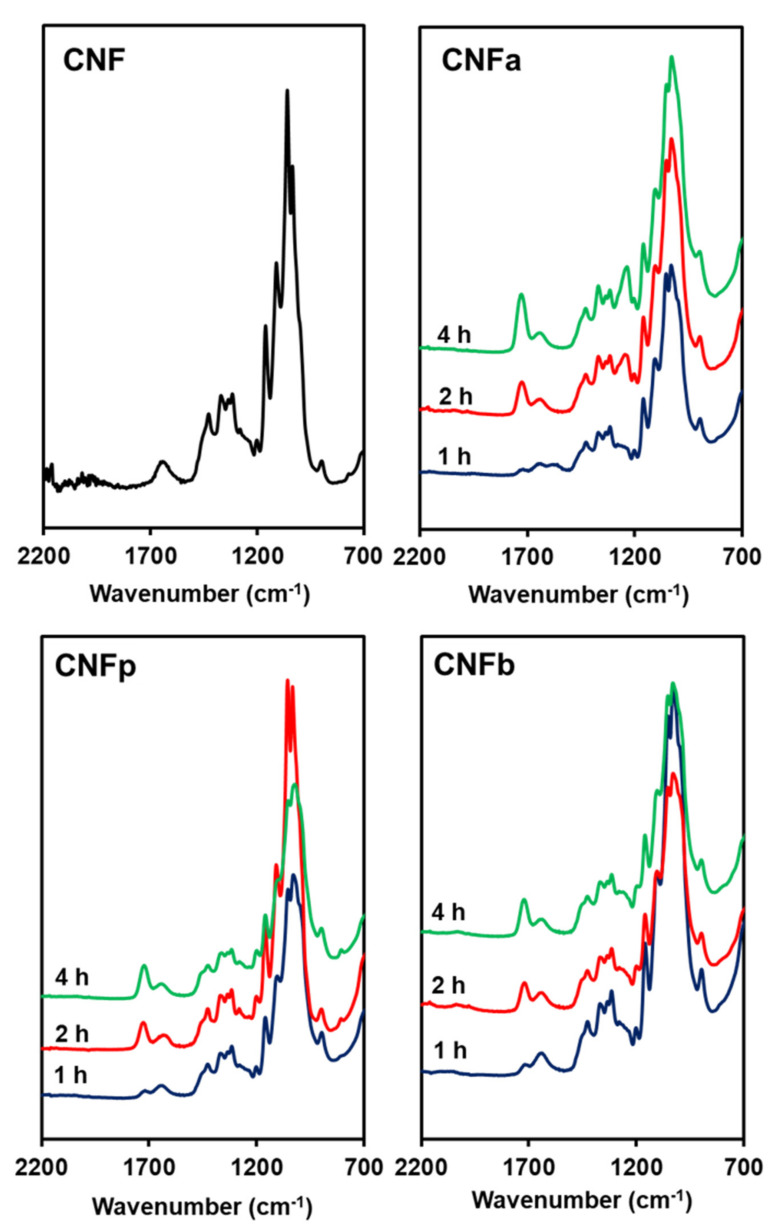
ATR-IR spectra of CNF, CNFa, CNFp, and CNFb.

**Figure 3 nanomaterials-11-00753-f003:**
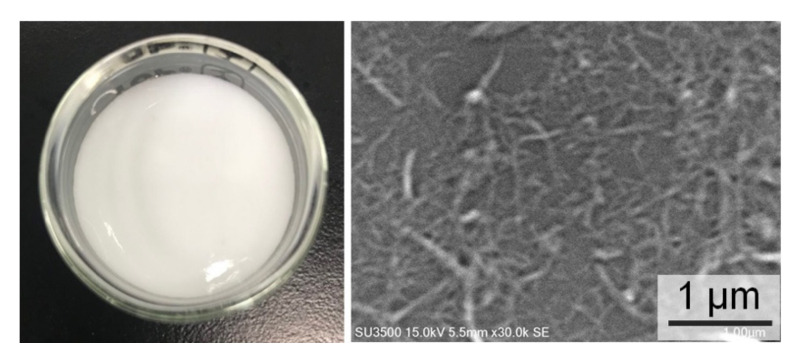
Suspension of CNF in water and its SEM image at ×30,000 magnification.

**Figure 4 nanomaterials-11-00753-f004:**
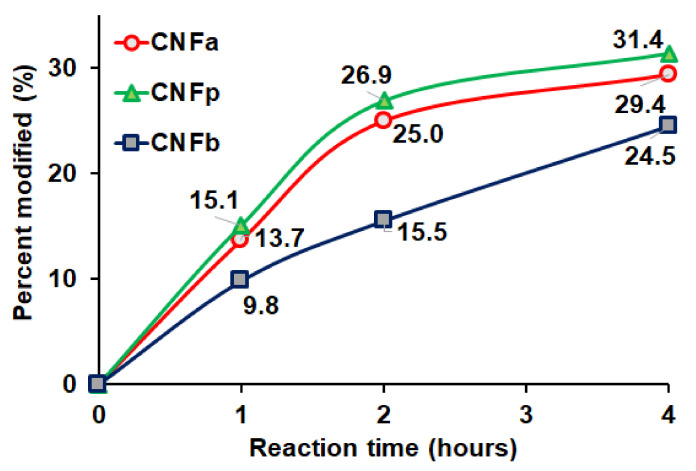
Percentage of CNF modified with respect to the reaction time for CNFa, CNFp, and CNFb.

**Figure 5 nanomaterials-11-00753-f005:**
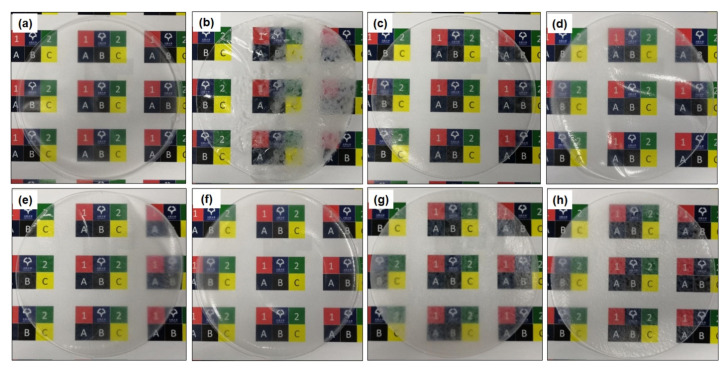
Photographs of the (**a**) neat PLA, (**b**) PLA/CNF, (**c**) PLA/CNFa2, (**d**) PLA/CNFa4, (**e**) PLA/CNFp2, (**f**) PLA/CNFp4, (**g**) PLA/CNFb2, and (**h**) PLA/CNFb4 composite films.

**Figure 6 nanomaterials-11-00753-f006:**
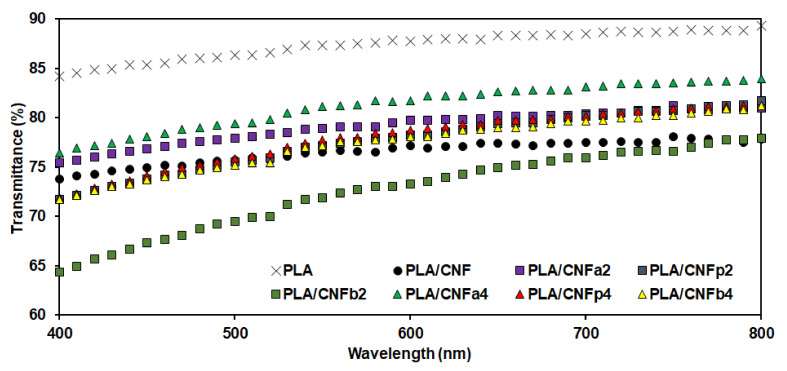
UV–vis transmittance of the PLA/m-CNF composite films (1 g of reactant).

**Figure 7 nanomaterials-11-00753-f007:**
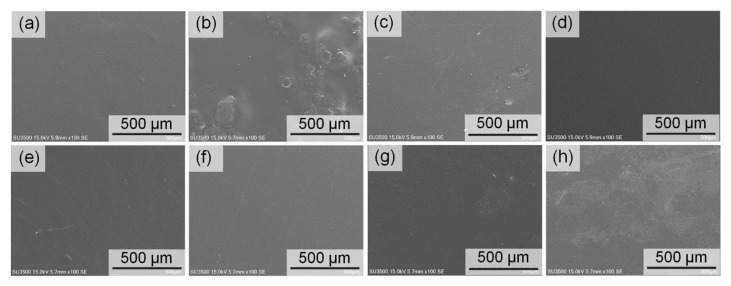
SEM images of the (**a**) neat PLA, (**b**) PLA/CNF, (**c**) PLA/CNFa2, (**d**) PLA/CNFa4, (**e**) PLA/CNFp2, (**f**) PLA/CNFp4, (**g**) PLA/CNFb2, and (**h**) PLA/CNFb4 composite films.

**Figure 8 nanomaterials-11-00753-f008:**
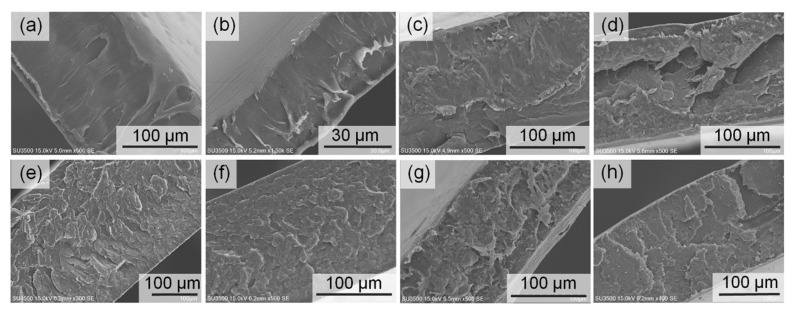
Cross-section images of the (**a**) neat PLA, (**b**) PLA/CNF, (**c**) PLA/CNFa2, (**d**) PLA/CNFa4, (**e**) PLA/CNFp2, (**f**) PLA/CNFp4, (**g**) PLA/CNFb2, and (**h**) PLA/CNFb4 composite films.

**Figure 9 nanomaterials-11-00753-f009:**
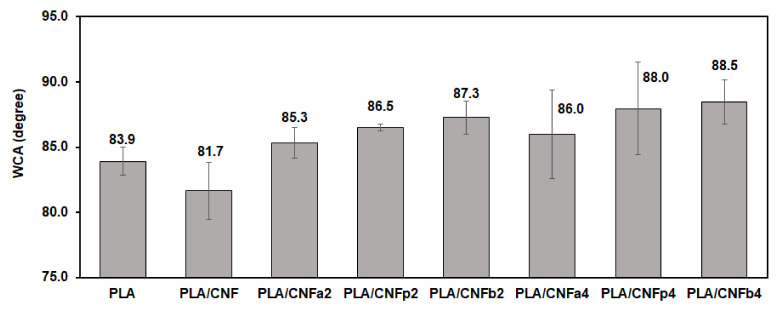
Wettability of the PLA composite films.

**Table 1 nanomaterials-11-00753-t001:** Degrees of substitutions (DS) values of the modified CNFs (m-CNFs) based on the reaction time.

Reaction Time (Hours)	Degree of Substitution (DS)		
	CNFa	CNFp	CNFb
1	0.41	0.45	0.29
2	0.75	0.81	0.47
4	0.88	0.94	0.74

**Table 2 nanomaterials-11-00753-t002:** Average thickness, transmittance, and transparency values of the PLA/m-CNF composite films.

Films	DS of Fillers	Thicknesses (μm)		UV–Vis Transmittance (%)		Transparency Values	
		1 g	2 g	1 g	2 g	1 g	2 g
Neat PLA	-	51	190	89.3	39.1	3.23	2.31
PLA/CNF	-	82	231	77.8	13.2	2.97	1.76
PLA/CNFa2	0.75	65	238	81.0	26.8	3.08	2.05
PLA/CNFp2	0.81	64	213	81.7	25.8	3.08	2.08
PLA/CNFb2	0.47	81	232	77.9	25.3	2.95	2.04
PLA/CNFa4	0.88	60	224	83.9	34.9	3.13	2.19
PLA/CNFp4	0.94	57	177	81.2	42.2	3.14	2.38
PLA/CNFb4	0.74	52	256	81.2	22.3	3.18	1.94

**Table 3 nanomaterials-11-00753-t003:** Average thicknesses and haze values of the PLA/m-CNF composite films.

Films	DS of Fillers	Thicknesses (μm)	Haze Values (%)
Neat PLA	-	51	6.8
PLA/CNF	-	82	35.5
PLA/CNFa2	0.75	65	19.5
PLA/CNFp2	0.81	64	17.8
PLA/CNFb2	0.47	81	30.0
PLA/CNFa4	0.88	60	20.1
PLA/CNFp4	0.94	57	17.5
PLA/CNFb4	0.74	52	19.0

**Table 4 nanomaterials-11-00753-t004:** Average tensile strength, Young’s modulus, tensile strain, and toughness of the PLA/m-CNF composite films.

Composite Films	Tensile Stress (MPa)	Young’s Modulus (GPa)	Tensile Strain (%)	Toughness (J/mm^3^)
Neat PLA	46.1 ± 4.5	1.24 ± 0.28	3.6 ± 0.5	1.54 ± 0.21
PLA/CNF	23.9 ± 5.8	1.00 ± 0.39	2.5 ± 0.4	0.25 ± 0.08
PLA/CNFa2	41.1 ± 0.2	1.41 ± 0.57	3.3 ± 1.4	0.49 ± 0.03
PLA/CNFp2	49.4 ± 3.0	1.75 ± 0.08	3.0 ± 0.4	0.80 ± 0.14
PLA/CNFb2	32.5 ± 1.4	1.23 ± 0.17	2.4 ± 0.3	0.90 ± 0.59
PLA/CNFa4	50.5 ± 2.1	1.07 ± 0.10	3.9 ± 0.5	1.65 ± 0.01
PLA/CNFp4	53.0 ± 2.4	1.74 ± 0.09	3.5 ± 0.1	1.31 ± 0.21
PLA/CNFb4	36.3 ± 0.1	1.08 ± 0.21	3.2 ± 0.2	0.81 ± 0.18

**Table 5 nanomaterials-11-00753-t005:** The degree of crystallinity for PLA/m-CNF composite films.

Films	Degree of Crystallinity, X_c_ (%)
Neat PLA	39.6
PLA/CNF	16.2
PLA/CNFa2	32.4
PLA/CNFp2	28.2
PLA/CNFb2	28.1
PLA/CNFa4	30.7
PLA/CNFp4	33.1
PLA/CNFb4	28.6

## Data Availability

The data presented in this study are available within the article or Appendix A.

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
