# Peer review of "Effects of Acid-Anhydride-Modified Cellulose Nanofiber on Poly(Lactic Acid) Composite Films"

_nanomaterials, 2021, doi:10.3390/nano11030753_

Round 1
Reviewer 1 Report
Cellulose nanofibres were modified by three chemicals to improve their interaction with PLA. The properties of PLA/cellulose composite were examined to find the best filler for PLA.
The surface of various composites was observed and compared by SEM. However, it would be better to compare their cross section after freeze cracking.
Author Response
Comments:
Cellulose nanofibres were modified by three chemicals to improve their interaction with PLA. The properties of PLA/cellulose composite were examined to find the best filler for PLA.
The surface of various composites was observed and compared by SEM. However, it would be better to compare their cross section after freeze cracking.
Authors’ response:
Thank you for the feedback and comments. As suggested, we observed the freeze cracking morphologies of the cross-section. In section 3.6 (Morphologies of PLA/m-CNF composite films), we added a new paragraph along with the cross-section figures (Figure 8) to explain the observation. By the addition of this figure, we also move the original Figure 8 into Figure 9.
(Line 333-341)
Reviewer 2 Report
Jamaluddin et al. in this work modified the hydroxyl group of cellulose (CNF) to the acyl group by reaction with acid anhydrides. The modified NCFs were mixed with PLA and the effects of the different acid anhydrides with different degrees of substitutions on the transparency, surface morphology, mechanical, and wettability properties of the composites were investigated. The compatibility between PLA and the modified CNF was improved and the propionylated CNF (CNFp) shows the best performance. The results of this work can be a good reference in the field of CNF applications. However, in addition to just reporting the results, there should be more discussions on the mechanisms. My comments are as follows.
- In Figure 4, CNFp showed a higher acylation rate than CNFa and CNFb. The reason should be discussed.
- In addition to the aggregation of CNFs, another factor that affects the transparency and mechanical properties of the composites is the crystallinity of PLA. The use of DSC to determine the relative crystallinity of PLA affected by the m-CNFs is suggested.
- For 2 g of the reactant (thicker films), I am wondering why the haze transmittance was not reliable and some bizarre results were obtained. If this is true, what is the reason to show these data?
- On Page 18, in the sentence “It seems that the propionyl group substituted on the hydroxyl group of CNF tends to shield and localize CNFb,” the “propionyl” should be corrected to “butyl.”
- The reason for CNFb4 (DS = 0.74) to form relatively large agglomerates as compared to other m-CNFs should be explained.
- The authors state that PLA/m-CNF composite films showed good tensile strength and modulus, their toughness was relatively poor because of the stiff backbone chain of PLA. How can the stiff backbone of PLA cause the poor toughness?
- The original data of the mechanical properties are suggested to show in the supplementary materials.
Author Response
Comments:
Jamaluddin et al. in this work modified the hydroxyl group of cellulose (CNF) to the acyl group by reaction with acid anhydrides. The modified NCFs were mixed with PLA and the effects of the different acid anhydrides with different degrees of substitutions on the transparency, surface morphology, mechanical, and wettability properties of the composites were investigated. The compatibility between PLA and the modified CNF was improved and the propionylated CNF (CNFp) shows the best performance. The results of this work can be a good reference in the field of CNF applications. However, in addition to just reporting the results, there should be more discussions on the mechanisms. My comments are as follows.
Authors’ response:
Thank you for the feedback and comments.
In Figure 4, CNFp showed a higher acylation rate than CNFa and CNFb. The reason should be discussed.
Authors’ response:
Thank you for pointing this out. We added a sentence to discuss the reason in section 3.2 (Degree of substitution) along with a newly added reference.
“Longer alkyl-chain of the anhydride (BA) leads to a slower rate reaction due to steric effect and lower surface area compared to AA and PA [29].”
(Line 214-215)
In addition to the aggregation of CNFs, another factor that affects the transparency and mechanical properties of the composites is the crystallinity of PLA. The use of DSC to determine the relative crystallinity of PLA affected by the m-CNFs is suggested.
Authors’ response:
Thank you for the suggestion. We characterized the thermal properties of our films by DSC analysis and calculated their degree of crystallinity. We added a new section 3.7 (Crystallinity of the PLA/m-CNF composite films) to explain the results. Along with that, we added sentences in characterization for the DSC measurement method and DSC thermograms in Supplementary Materials (Figure S3). Original section 3.7 was arranged to 3.8 (Wettability of the PLA/m-CNF composite films)
(Line 171-180 and 373-385)
For 2 g of the reactant (thicker films), I am wondering why the haze transmittance was not reliable and some bizarre results were obtained. If this is true, what is the reason to show these data?
Authors’ response:
Thank you for the comments. In section 3.5 (Haze transmittance of PLA/m-CNF composite film), we agreed with you and removed all explanations about the sample with 2 g of the reactant, including the results in Table 3 due to the unreliable results.
(Line 284-296)
On Page 18, in the sentence “It seems that the propionyl group substituted on the hydroxyl group of CNF tends to shield and localize CNFb,” the “propionyl” should be corrected to “butyl.”
Authors’ response:
Thank you for pointing out our mistake. In the same sentence, we corrected the term “propionyl” into “butyryl”.
(Line 319)
The reason for CNFb4 (DS = 0.74) to form relatively large agglomerates as compared to other m-CNFs should be explained.
Authors’ response:
Thank you for the comments. We added a sentence after the mentioned above statement in section 3.6 (Morphologies of the PLA/m-CNF composite films) to explain the observation.
“It appears that the higher number of the hydroxyl groups in CNFb4 (due to lower DS than CNFa4 and CNFp4) formed opposite interaction between unmodified and modified parts of CNFb4, hence, larger agglomerates were formed during dispersion in chloroform.”
(Line 325-328)
The authors state that PLA/m-CNF composite films showed good tensile strength and modulus, their toughness was relatively poor because of the stiff backbone chain of PLA. How can the stiff backbone of PLA cause the poor toughness?
Authors’ response:
Thank you for pointing this out. We realized that the sentence was confusing and might be incorrect. Therefore, we rephrase the sentence by replacing “… relatively poor because of the stiff backbone chain of PLA” with “… relatively poor, the same as general PLA”. We also changed reference 31 (arrangement after the correction) according to the change of the statement/sentence.
(Line 369)
The original data of the mechanical properties are suggested to show in the supplementary materials.
Author response:
Thank you for the suggestion. We added stress-strain curves of PLA/m-CNF composite films in the supplementary materials, which were used for obtaining Young’s modulus and toughnesses of the films. Along with the curves, we mentioned this additional figure in the main text in section 3.7 (Mechanical properties of the PLA/m-CNF composite films) as Figure S2.
(Line 367)
Reviewer 3 Report
The paper is very interesting, well organized and accurate, so I suggest the publication in the present form.
Author Response
Comments:
The paper is very interesting, well organized and accurate, so I suggest the publication in the present form.
Authors’ response:
We highly appreciated your comments. Thank you for the recommendation!
Round 2
Reviewer 1 Report
SEM of cross section was added.
Reviewer 2 Report
The authors have well responded to my comments. I thus recommend publication.